# Exome-wide association study reveals largely distinct gene sets underlying specific resistance to dengue virus types 1 and 3 in *Aedes aegypti*

**Laura B. Dickson**[1], **Sarah H. Merkling**[1], **Mathieu Gautier**[2], **Amine Ghozlane**[3], **Davy Jiolle**[1,4,5], **Christophe Paupy**[4,5], **Diego Ayala**[4,5], **Isabelle Moltini-Conclois**[1,4], **Albin Fontaine**[1¤a¤b], **Louis Lambrechts**[1]*

**1** Insect-Virus Interactions Unit, Institut Pasteur, UMR2000, CNRS, Paris, France, **2** CBGP, INRAE, CIRAD, IRD, Montpellier SupAgro, Univ. Montpellier, Montpellier, France, **3** Hub de Bioinformatique et Biostatistique–Département Biologie Computationnelle, Institut Pasteur, USR 3756 CNRS, Paris, France, **4** MIVEGEC, Univ. Montpellier, IRD, CNRS, Montpellier, France, **5** Centre Interdisciplinaire de Recherches Médicales de Franceville, Franceville, Gabon

☯ These authors contributed equally to this work.
¤a Current address: Unité de Parasitologie et Entomologie, Département des Maladies Infectieuses, Institut de Recherche Biomédicale des Armées (IRBA), Marseille, France
¤b Current address: Aix Marseille Université, IRD, AP-HM, SSA, UMR Vecteurs-Infections Tropicales et Méditerranéennes (VITROME), IHU-Méditerranée Infection, Marseille, France
* louis.lambrechts@pasteur.fr

**Data Availability Statement:** Raw sequencing data were deposited to the European Nucleotide Archive repository under accession number PRJEB36810.

## Abstract

Although specific interactions between host and pathogen genotypes have been well documented in invertebrates, the identification of host genes involved in discriminating pathogen genotypes remains a challenge. In the mosquito *Aedes aegypti*, the main dengue virus (DENV) vector worldwide, statistical associations between host genetic markers and DENV types or strains were previously detected, but the host genes underlying this genetic specificity have not been identified. In particular, it is unknown whether DENV type- or strain-specific resistance relies on allelic variants of the same genes or on distinct gene sets. Here, we investigated the genetic architecture of DENV resistance in a population of *Ae. aegypti* from Bakoumba, Gabon, which displays a stronger resistance phenotype to DENV type 1 (DENV-1) than to DENV type 3 (DENV-3) infection. Following experimental exposure to either DENV-1 or DENV-3, we sequenced the exomes of large phenotypic pools of mosquitoes that are either resistant or susceptible to each DENV type. Using variation in single-nucleotide polymorphism (SNP) frequencies among the pools, we computed empirical *p* values based on average gene scores adjusted for the differences in SNP counts, to identify genes associated with infection in a DENV type-specific manner. Among the top 5% most significant genes, 263 genes were significantly associated with resistance to both DENV-1 and DENV-3, 287 genes were only associated with DENV-1 resistance and 290 were only associated with DENV-3 resistance. The shared significant genes were enriched in genes with ATP binding activity and sulfur compound transmembrane transporter activity, whereas the genes uniquely associated with DENV-3 resistance were enriched in genes with zinc ion binding activity. Together, these results indicate that specific resistance to different DENV

**Funding:** This work was funded by Agence Nationale de la Recherche (grants ANR-17-ERC2-0016-01 and ANR-18-CE35-0003-01 to LL), the French Government's Investissement d'Avenir program Laboratoire d'Excellence Integrative Biology of Emerging Infectious Diseases (grant ANR-10-LABX-62-IBEID to LL), and the City of Paris Emergence(s) program in Biomedical Research (to LL). The funders had no role in study design, data collection and analysis, decision to publish, or preparation of the manuscript.

**Competing interests:** The authors have declared that no competing interests exist.

types relies on largely non-overlapping sets of genes in this *Ae. aegypti* population and pave the way for further mechanistic studies.

## Author summary

Compatibility between hosts and pathogens is often genetically specific in invertebrates but host genes underlying this genetic specificity have not been elucidated. We investigated the genetic architecture of dengue virus type-specific resistance in the mosquito vector *Aedes aegypti*. We used a natural population of *Ae. aegypti* from Bakoumba, Gabon, which is differentially resistant to dengue virus type 1 and dengue virus type 3. We surveyed genetic variation in protein-coding regions of the mosquito genome and compared the frequency of genetic polymorphisms between groups of mosquitoes that are either resistant or susceptible to each dengue virus type. We found that the *Ae. aegypti* genes associated with resistance to dengue virus type 1 or dengue virus type 3 were largely non-overlapping. This finding indicates that different sets of host genes, rather than different variants of the same genes, confer pathogen-specific resistance in this population. This study is an important step towards identification of mechanisms underlying the genetic specificity of invertebrate host-pathogen interactions.

## Introduction

In many invertebrate host-pathogen systems, infection success depends on the specific pairing of host and pathogen genotypes [1]. Such genotype-by-genotype (G x G) interactions have been observed, for example, between crustaceans and bacteria [2], bumblebees and intestinal trypanosomes [3], nematodes and bacteria [4], anopheline mosquitoes and malaria parasites [5, 6] and butterflies and protozoan parasites [7]. In some instances, G x G interactions can result in extreme levels of host-pathogen specificity [8].

Understanding this genetic specificity in invertebrate host-pathogen systems has generated great enthusiasm because it may uncover new facets of invertebrate immunity [9, 10]. Although invertebrates lack the adaptive immunity of vertebrates, their immune system can generate a considerable diversity of immune receptors, revealing an unsuspected molecular complexity [11, 12]. Alternatively, G x G interactions could be mediated by variation in the host microbiota [13] or any other interplay between host and pathogen genomes. G x G interactions can be detected at the level of gene expression [14] and mapped to physical locations in the host genome [15–17]. Accumulating observations on G x G interactions have led to some controversy [18–20] because they challenge the prevailing view that invertebrate defense against pathogens relies on broad-spectrum recognition and effector mechanisms. A central aspect of the controversy is that the molecular mechanisms underlying G x G interactions between invertebrate hosts and pathogens are yet to be elucidated.

We previously documented significant G x G interactions between dengue virus (DENV) and its main mosquito vector *Aedes aegypti* [21, 22]. Both DENV and *Ae. aegypti* are genetically diverse in nature. DENV exists worldwide as four genetic types (DENV-1, DENV-2, DENV-3, and DENV-4) that loosely cluster antigenically [23] and are often referred to as serotypes. The mosquito *Ae. aegypti* consists of two major genetic subspecies that contain substantial genetic diversity [24]. In the last few decades, DENV has become a major public health threat worldwide with an estimated 390 million human infections per year [25]. The lack of an

efficient vaccine and the failure of insecticide-based methods to control mosquito populations on the long term have stimulated basic research to develop novel vector control strategies [26]. One of these strategies aims at rendering mosquitoes resistant to pathogen infection [27]. The premise of this strategy makes it critical to understand G x G interactions because engineered mosquito resistance should be effective against all possible pathogen genotypes.

Dissecting the genetic basis of G x G interactions between DENV and *Ae. aegypti* was previously hampered by the large size of the *Ae. aegypti* genome and the paucity of genetic markers available [28]. Although G x G interactions between DENV and *Ae. aegypti* were statistically assigned to individual mosquito genetic markers [15, 17], the specific mosquito genes underlying these quantitative trait loci (QTL) have not been identified. In particular, it is unknown whether DENV type- or strain-specific resistance relies on allelic variants of the same genes or on distinct gene sets.

Here, we combined a natural phenotype of DENV type-specific resistance in *Ae. aegypti* and the power of high-throughput sequencing to provide insights into the genetic architecture of G x G interactions in this system. We previously established a laboratory colony from a wild *Ae. aegypti* population in Bakoumba, Gabon that is differentially susceptible to DENV types. Using dose-response experiments, we found the Bakoumba population to be strongly resistant to DENV-1 and only moderately resistant to DENV-3 infection. We took advantage of this field-derived colony to investigate the genetic basis for discriminating between different DENV types. We used a modified genome-wide association study design, in which replicate pools of individuals with contrasted phenotypes are pooled by phenotype and analyzed for differences in allele frequencies [29, 30]. We compared the genetic basis of Bakoumba resistance to DENV-1 and DENV-3 from extremes of the phenotypic distribution (i.e., resistant = uninfected at a high virus dose vs. susceptible = infected at a low virus dose). We estimated and compared allele frequencies among the phenotypic pools using the contrast statistic recently implemented in the software BayPass [31, 32]. Because the genome of *Ae. aegypti* is large and repetitive, we performed exome sequencing as a cost-effective genotyping method targeting the exons of all protein-coding genes [33], as was previously implemented in *Ae. aegypti* [34–36]. We combined SNP-specific contrasts of allele frequency to compute gene-wide scores reflecting the statistical significance of association with DENV-1 and/or DENV-3 infection. The results of our exome-wide association study (EWAS) show that largely distinct gene sets underlie specific resistance to DENV-1 and DENV-3 infection in this *Ae. aegypti* population.

## Results

### Dose-response experiments

We initially compared the infection dose response of two outbred *Ae. aegypti* colonies from Cairns (Australia) and Bakoumba (Gabon) following oral exposure to either DENV-1 or DENV-3. In two separate experiments, female mosquitoes from both colonies were offered artificial blood meals containing one of three increasing infectious doses ranging from $1.0 \times 10^4$ to $3.3 \times 10^7$ FFUs/mL for DENV-1 and from $1.0 \times 10^4$ to $3.9 \times 10^7$ FFUs/mL for DENV-3 (Fig 1A). Ten days after virus exposure, infection status was determined by RT-PCR for 24–32 (mean 30.8; median 32) blood-fed females for each combination of experiment, population, virus and dose. In total, 366 individual females were tested for DENV-1 and 374 individual females were tested for DENV-3. As expected, infectious dose was a significant predictor of infection prevalence (Table 1). In addition, we observed a strong pattern of G x G interactions, whereby the Bakoumba population was significantly more resistant to DENV-1 than the three other population-virus combinations (Fig 1A). This G x G interaction was statistically supported by a significant effect ($p = 0.0013$) of the population x virus interaction on the infection

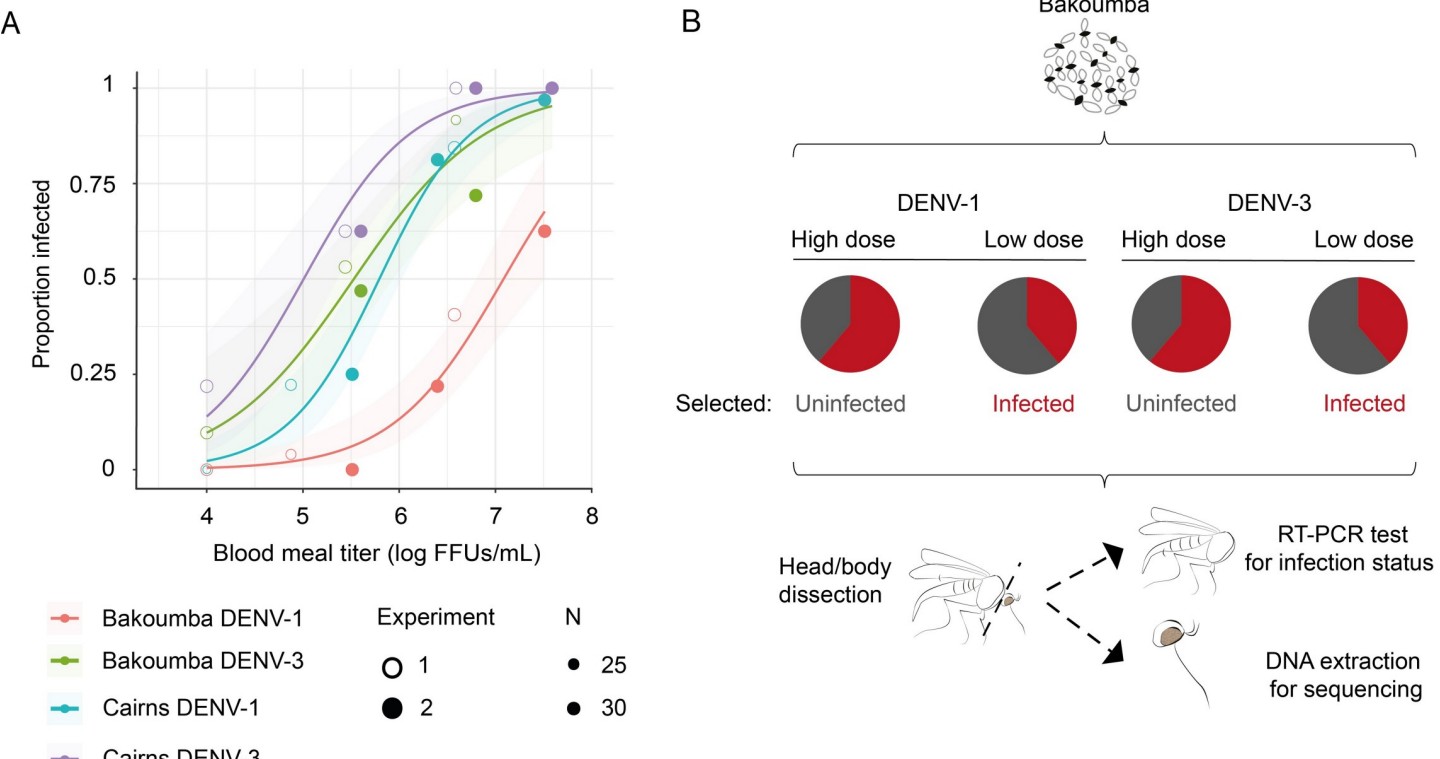

**Fig 1. Investigating the DENV type-specific resistance phenotype of Bakoumba mosquitoes.** (**A**) Dose-response curves of *Ae. aegypti* colonies from Bakoumba (Gabon) or Cairns (Australia) challenged with DENV-1 and DENV-3. The percentage of DENV-infected mosquitoes at 10 days post exposure is shown as a function of the blood meal titer in $\log_{10}$-transformed focus-forming units (FFUs)/mL. The data was obtained in two separate dose-response experiments with three doses for each virus. The three doses of experiment 1 are covering a lower range than the three doses of experiment 2. Curves are logistic regressions of the data with their 95% confidence intervals indicated by shaded bands. Note that one data point (Bakoumba DENV-3 experiment 2) is masked due to the overlap with another value at 100% (Cairns DENV-3 experiment 2). (**B**) Experimental design for the exome-wide association analysis (EWAS) of DENV type-specific resistance of Bakoumba mosquitoes. Female mosquitoes of the Bakoumba population were exposed to low and high infectious doses of DENV-1 and DENV-3, expected to result in about one third and two thirds of infected mosquitoes, respectively. Bodies were used to screen for infection status by RT-PCR, and heads were collected for DNA extraction and exome sequencing. The EWAS analysis compared pools of individuals that were uninfected at the high dose (i.e., resistant) versus individuals infected at the low dose (i.e., susceptible) for each DENV type.

phenotype (Table 1). Based on the regression curves of the combined dose responses, we estimated that the 50% oral infectious doses ($OID_{50}$) of the Bakoumba population were 7.08 $\log_{10}$ FFUs/mL (95% confidence interval: 6.84–7.44) for DENV-1 and 5.54 $\log_{10}$ FFUs/mL (95% CI:

**Table 1. Test statistics of the DENV dose-response analysis.** Infection prevalence was analyzed by logistic regression as a function of replicate experiment, dose, population, virus, and all their interactions. The table shows the minimal adequate model after sequentially removing non-significant effects. Oral infectious dose (blood meal titer) was $\log_{10}$-transformed prior to the analysis.

| Source | Df | L-R $\chi^2$ | P value |
|---|---|---|---|
| Experiment | 1 | 9.882 | 0.0017 |
| Dose | 1 | 290.8 | <0.0001 |
| Population | 1 | 64.79 | <0.0001 |
| Virus | 1 | 87.90 | <0.0001 |
| Population x Virus | 1 | 10.39 | 0.0013 |

Df = degrees of freedom; L-R = likelihood-ratio.

5.23–5.80) for DENV-3. The $OID_{50}$ estimates of the Cairns population were 5.80 $log_{10}$ FFUs/mL (95% CI: 5.58–6.00) for DENV-1 and 5.01 $log_{10}$ FFUs/mL (95% CI: 4.68–5.26) for DENV-3. The dose-response analysis showed that the Bakoumba population was intrinsically more resistant to DENV-1 than to DENV-3 infection, although the DENV-1 and DENV-3 isolates were equally infectious to the mosquito population from Cairns. In the rest of the study, we took advantage of this natural resistance phenotype to dissect the genetic architecture of DENV type-specific resistance in *Ae. aegypti*. A direct comparison between the Bakoumba and the Cairns populations would be impractical because their genetic differentiation would likely mask any loci underlying DENV resistance that are shared or divergent between them. Indeed, the Bakoumba population presumably belongs to the African subspecies, *Ae. aegypti formosus*, whereas the Cairns population presumably belongs to the globally invasive subspecies, *Ae. aegypti aegypti*, therefore the two populations are expected to be genetically divergent [24]. In contrast, the comparison between phenotypically resistant or susceptible individuals within the Bakoumba population allowed us to control for the genetic background.

## Exome-wide association study

We reasoned that if Bakoumba mosquitoes were intrinsically more resistant to DENV-1 than to DENV-3 infection, the genetic factors underlying resistance to each DENV type would be distinct. Our study was designed to determine whether DENV type-specific resistance in the Bakoumba population relied on different alleles of the same genes (i.e., if a DENV-1 resistance allele also confers susceptibility to DENV-3) or on genetic variation at distinct gene sets (i.e., if DENV-1 resistance and susceptibility alleles do not influence DENV-3 infection). To identify allelic variants associated with both DENV-1 and DENV-3 resistance or susceptibility, we sequenced the exome of four phenotypic pools of Bakoumba mosquitoes (Fig 1B): DENV-1 resistant (i.e., uninfected after exposure to a high DENV-1 dose), DENV-1 susceptible (i.e., infected after exposure to a low DENV-1 dose), DENV-3 resistant (i.e., uninfected after exposure to a high DENV-3 dose), and DENV-3 susceptible (i.e., infected after exposure to a low DENV-3 dose). Low and high infectious doses were adjusted to result in approximately one third and two thirds of infected mosquitoes, respectively. Due to the difference in the dose responses (Fig 1A), low and high infectious doses were different for each DENV type (Table 2). DNA was extracted from the head tissues of each individual mosquito whereas the infection phenotype was determined by RT-PCR testing of the head-less mosquito body 10 days post infectious blood meal. The phenotypic screen was performed in three replicate experiments.

## Association analyses at the single-nucleotide level

The bodies of 668, 690 and 680 females was individually tested for DENV infection by RT-PCR in replicate experiments 1, 2 and 3, respectively (Table 3). Based on their resistant or susceptible phenotype, 182, 174 and 176 females were selected for DNA extraction from experiments 1, 2 and 3, respectively. DNA extracted from the head of individual females was combined into 12 phenotypic pools (3 experiments x 2 DENV types x 2 phenotypes) of 30–48 individuals (mean 44.3) to prepare 12 libraries for exome sequencing (Table 3). Individual DNA concentrations were adjusted prior to pooling so that each individual contributed the same amount of DNA to the library. Across all three chromosomes, a total of approximately 1.4 million SNPs were identified with an average sequencing depth ranging from 100.6X to 176.4X (mean 132.7X) among the 12 libraries. We first performed association analyses at the level of individual SNPs based on allele frequency differences between DENV-1 resistant and susceptible pools (DENV-1 comparison) and between DENV-3 resistant and susceptible pools

**Table 2. Infectious doses used for the EWAS phenotypic screen.** The table shows the measured titer of DENV-1 and DENV-3 infectious blood meals for each of the three experimental replicates of the exome-wide association study in the Bakoumba population.

| Experiment | Virus | Dose | Titer (FFUs/mL) |
|---|---|---|---|
| Exp. 1 | DENV-1 | High | $1.00 \times 10^7$ |
| | | Low | $1.26 \times 10^6$ |
| | DENV-3 | High | $7.75 \times 10^5$ |
| | | Low | $7.25 \times 10^4$ |
| Exp. 2 | DENV-1 | High | $1.10 \times 10^7$ |
| | | Low | $8.75 \times 10^5$ |
| | DENV-3 | High | $9.13 \times 10^5$ |
| | | Low | $4.88 \times 10^4$ |
| Exp. 3 | DENV-1 | High | $8.63 \times 10^6$ |
| | | Low | $9.50 \times 10^5$ |
| | DENV-3 | High | $4.75 \times 10^5$ |
| | | Low | $5.25 \times 10^4$ |

FFU = focus-forming unit

(DENV-3 comparison). For both the DENV-1 and the DENV-3 comparisons, the largest number of significant SNPs was detected on chromosome 1 but each DENV type displayed a distinct profile (S1A Fig and S1B Fig). Among the top 0.001% most significant SNPs for each comparison, 5 SNPs were shared between the DENV-1 and DENV-3 comparisons, 142 were uniquely associated with DENV-1 infection status, and 146 were uniquely associated with DENV-3 infection status (S1 Table; S1C Fig). For the 5 shared SNPs, allelic effects were in the same direction (i.e., the same allele was associated with both DENV-1 and DENV-3 resistance).

## Association analyses at the gene level

To identify candidate genes associated with DENV-1 and/or DENV-3 infection status, we performed a second series of association analyses based on an average score calculated for each

**Table 3. Summary of mosquitoes analyzed for each experimental replicate of the EWAS phenotypic screen.** The table shows the number of Bakoumba mosquitoes that tested DENV-positive or DENV-negative by RT-PCR (S2 Fig) in each of the experimental conditions of the screen. Cells highlighted in yellow are experimental conditions used as opposite phenotypes for the exome-wide association study. The number shown in brackets represents the number of mosquitoes selected for DNA extraction and subsequent sequencing.

| Experiment | Virus | Dose | Infection prevalence | Number infected | Number uninfected | Total analyzed |
|---|---|---|---|---|---|---|
| Exp. 1 | DENV-3 | Low | 27% | 42 (41) | 116 | 158 |
| | DENV-3 | High | 59% | 83 | 58 (48) | 141 |
| | DENV-1 | Low | 38% | 70 (48) | 116 | 186 |
| | DENV-1 | High | 75% | 138 | 45 (45) | 183 |
| Exp. 2 | DENV-3 | Low | 34% | 64 (48) | 124 | 188 |
| | DENV-3 | High | 65% | 117 | 62 (48) | 179 |
| | DENV-1 | Low | 36% | 63 (48) | 110 | 173 |
| | DENV-1 | High | 80% | 120 | 30 (30) | 150 |
| Exp. 3 | DENV-3 | Low | 26% | 47 (45) | 135 | 182 |
| | DENV-3 | High | 67% | 117 | 57 (48) | 174 |
| | DENV-1 | Low | 57% | 85 (48) | 63 | 148 |
| | DENV-1 | High | 80% | 141 | 35 (35) | 176 |

annotated gene. An empirical *p* value was computed from the average gene scores adjusted for differences in SNP count per gene. Note that although the exome capture probes were designed using the AaegL3 genome build [28] because it was the only one available at the time, our gene-based analysis was performed using the more recent AaegL5 genome build [17], which is significantly better annotated. Of the 19,763 genes annotated in the AaegL5 reference sequence, our exome capture probes covered 11,076 genes. Overall, the chromosomal distribution of empirical *p* values computed for each gene and each DENV type (Fig 2) was consistent with the pattern observed with the SNP-based analysis (S1 Fig). The most significant genes were detected on chromosome 1 around the centromere for both DENV types. Although there are regions of similarity between the DENV-1 and DENV-3 comparisons, the genetic architecture underlying DENV-1 and DENV-3 infection phenotypes is markedly distinct (Fig 2). Among the top 5% most significant candidate genes for each DENV type, 287 genes are uniquely associated with DENV-1 infection status, 290 are uniquely associated with DENV-3 infection status, and 263 genes are shared (S2 Table; Fig 3A).

Finally, we sought to determine whether the 5% most significant genes in the gene-based analysis belonged to different functional categories. We focused on the molecular mechanisms potentially underlying DENV type-specific resistance and accordingly we extracted gene ontology (GO) terms that exhibited a significant enrichment in the molecular function category. The 290 genes uniquely associated with resistance to DENV-3 infection were enriched in genes with zinc ion binding activity (Fig 3B). The 287 genes uniquely associated with resistance to DENV-1 were not significantly enriched in any particular molecular function. The 263 genes shared between the two comparisons were enriched in genes with ATP binding activity and sulfur compound transmembrane transporter activity (Fig 3B).

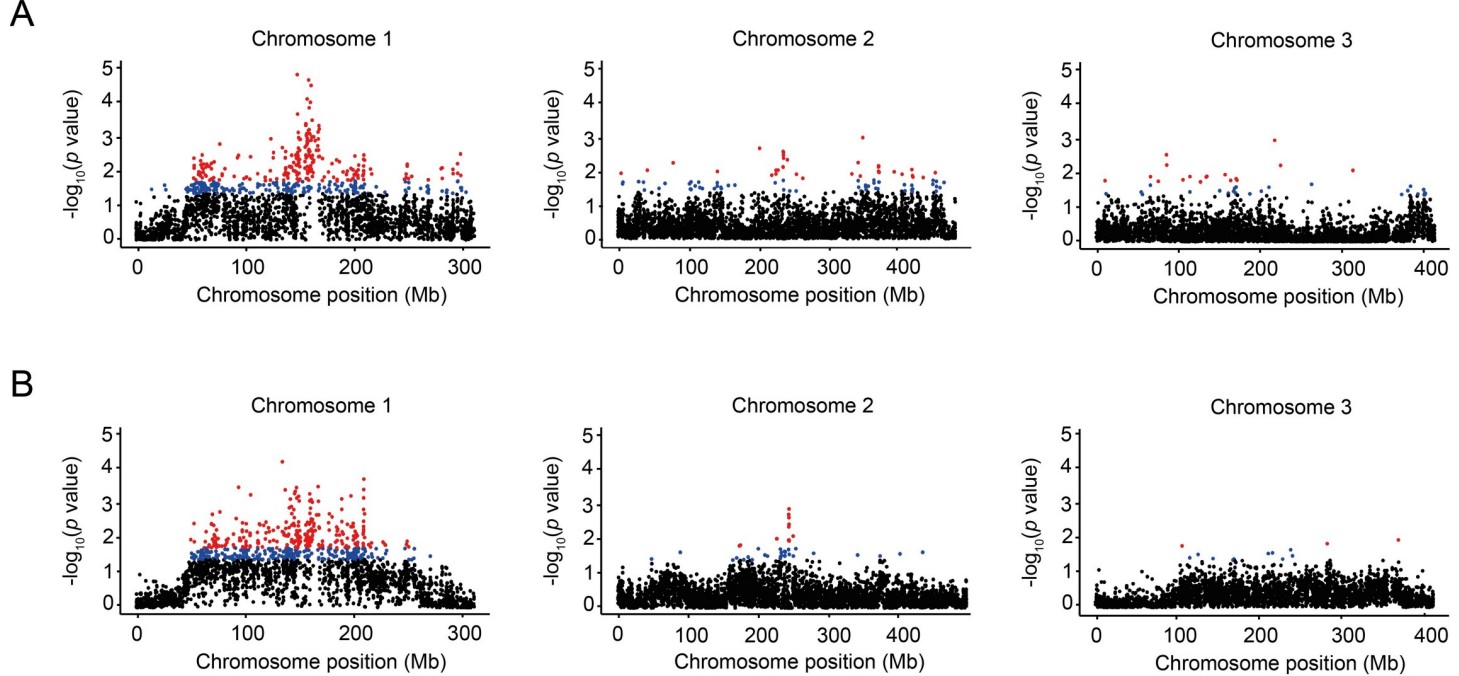

**Fig 2. Genes associated with DENV-1 and DENV-3 infection in the Bakoumba population.** **(A)** Manhattan plot of empirical *p* values derived from average gene scores of DENV-1 resistance, distributed along the three chromosomes. The gene scores reflect variation in SNP frequencies (adjusted for differences in SNP counts) between phenotypic pools of mosquitoes that are either resistant of susceptible to DENV-1. **(B)** Manhattan plot of empirical *p* values derived from average gene scores of DENV-3 resistance, distributed along the three chromosomes. The gene scores reflect variation in SNP frequencies (adjusted for differences in SNP counts) between phenotypic pools of mosquitoes that are either resistant of susceptible to DENV-3. Each dot represents a single gene and is colored according to the statistical significance of the genotype-phenotype association. Blue dots represent the lower 5% of *p* values and red dots represent the lower 2.5% *p* values.

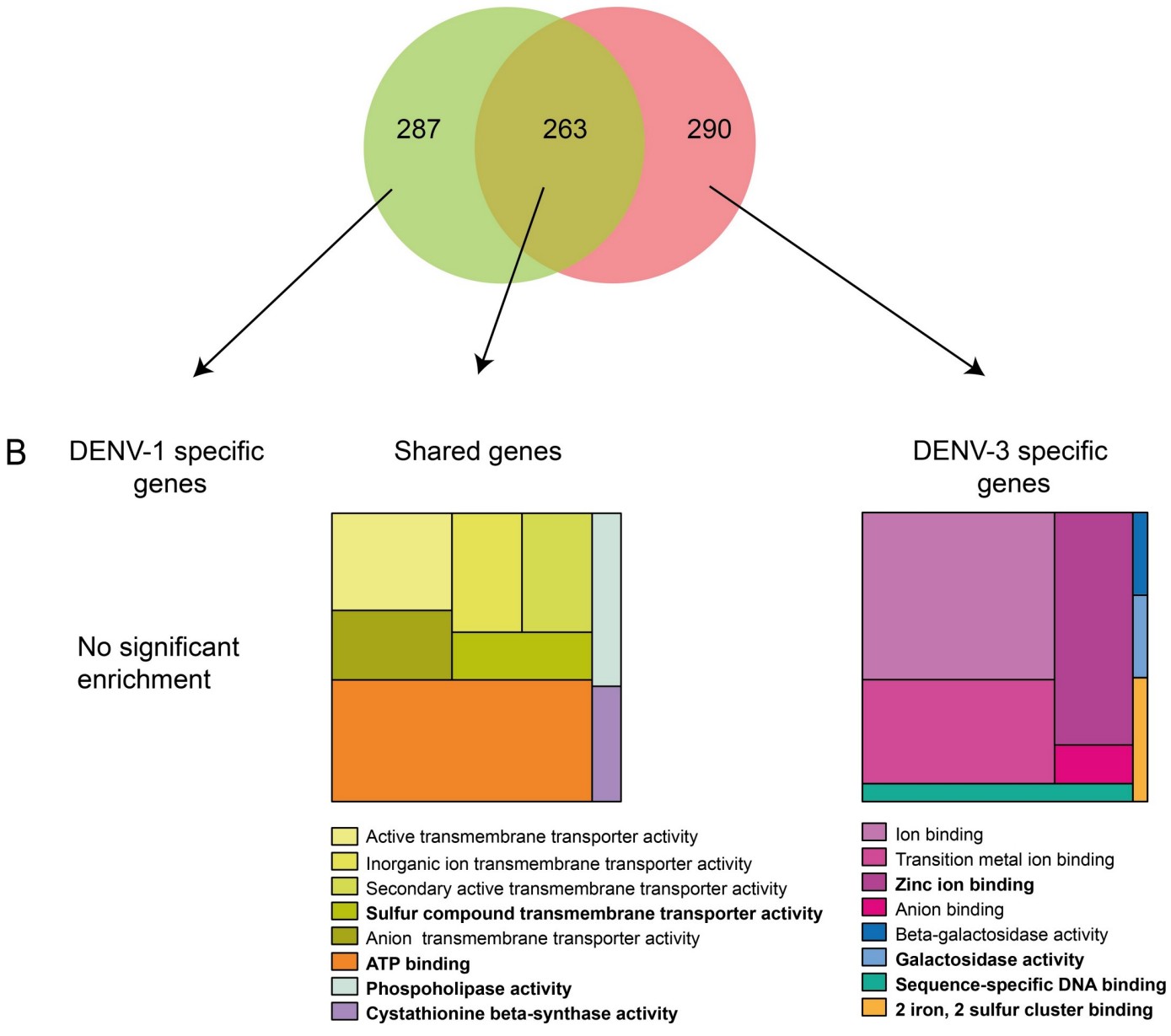

**Fig 3. Limited overlap between genes associated with DENV-3 or DENV-1 infection in the Bakoumba population.** (**A**) Venn diagram of the 5% most significant genes associated with resistance to DENV-1, resistance to DENV-3, or both. (**B**) TreeMap visualization of molecular function enrichment for the 5% most significant genes. Each first level category is represented by a large rectangle filled with nested rectangles corresponding to the GO term annotation. The size of the rectangle is proportional to the number of genes in the list out of the total number of genes with similar roles in the *Ae. aegypti* genome. Bold font represents a higher category level.

## Discussion

Our EWAS revealed that the DENV type-specific resistance phenotype displayed by an *Ae. aegypti* population from Bakoumba, Gabon reflects a distinct genetic architecture of resistance to DENV-1 and DENV-3 infection. The midgut infection phenotype we focused on is an important component of vector competence, which also depends on subsequent virus dissemination from the midgut and virus release in saliva [37]. We unveiled a set of SNPs and genes

that are significantly associated with DENV-1 and/or DENV-3 infection in the Bakoumba population, which open the way to future studies to functionally characterize the genetic basis of DENV type-specific resistance. The overlap between SNPs and genes associated with resistance to either DENV-1 or DENV-3 was only partial and demonstrated that DENV type-specific resistance relies on both shared and unique genes in this *Ae. aegypti* population. Although the mechanisms of DENV type-specific resistance in *Ae. aegypti* remain to be elucidated, this study is a leap forward to understand G x G specificity between *Ae. aegypti* and DENV at the molecular level.

G x G specificity between pathogens and their invertebrate hosts has been observed in many systems [1], but to date the molecular nature of this specificity is largely unknown. We previously detected G x G interactions between DENV and *Ae. aegypti* [21, 22] that could be statistically associated with QTL in the mosquito genome [15, 17]. However, the specific mosquito genes underlying these QTL have not been identified. One important question to elucidate the genetic basis of host-pathogen specificity is whether pathogen genotype-specific resistance relies on different alleles or different genes of the host. In the "matching-allele" model of host-pathogen interactions, a resistant allele against a given pathogen strain becomes a susceptible allele against another strain of the same pathogen [38]. This was observed, for example, by experimental viral evolution to specific mouse MHC genotypes [39] and by genetic crosses between genotypes of the crustacean *Daphnia magna* with different levels of resistance to the parasitic bacterium *Pasteuria ramosa* [40, 41]. In other host-pathogen systems, host resistance to different strains of the same pathogen relies on allelic variation at different genes. For example, genome-to-genome analysis of associations between human genetic variation and HIV-1 sequence found SNPs in different HLA class I genes [42]. Likewise, *Arabidopsis thaliana* SNPs exhibited genetic effects only when paired with certain *Xanthomonas arboricola* variants [43]. In the mosquito *Anopheles gambiae*, discrimination between human and rodent malaria species was shown to rely on different paralogs of the APL1 gene family [44]. Resistant and susceptible alleles of the TEP1 gene of *An. gambiae* explained resistance to some but not all strains of the human malaria parasite *Plasmodium falciparum* [45, 46].

We took advantage of the striking DENV type-specific phenotype of our Bakoumba *Ae. aegypti* colony to use it as a simplified model of G x G interactions that could be subjected to a genetic association study. Note that in this study, we focused on the host factors involved in the G x G interaction but the virus factors remain to be elucidated. We found that in the Bakoumba population, about half of the genes associated with resistance to DENV-1 infection did not confer resistance to DENV-3 infection and vice versa. Whether the genetic architecture of DENV type-specific resistance in the Bakoumba population is representative of G x G interactions between *Ae. aegypti* and DENV, and in other invertebrate host-pathogen systems in general, is unknown. Additional work will be necessary to extend these results to other mosquito populations, other DENV types and strains, and other host-pathogen systems. Moreover, our association study focused on the discovery of SNPs in exons, which represent only about 2% of the entire *Ae. aegypti* genome [17, 28]. It is possible that our exome-sequencing approach may have missed important genetic determinants underlying DENV type-specific resistance because they are not readily detected by SNPs (e.g., structural variants) and/or they occur in introns or intergenic regions. As sequencing technologies become more affordable, whole-genome sequencing will allow more comprehensive surveys of genetic variation in the *Ae. aegypti* genome. Nevertheless, detection of different exome variants associated with resistance to DENV-1 and resistance to DENV-3 is sufficient to support our conclusion that their respective genetic architecture is distinct.

It is worth noting that regardless of the biases introduced by exome sequencing, these biases are consistent across experimental conditions. In other words, comparing the genetic basis of

resistance to DENV-1 infection to the genetic basis of resistance to DENV-3 infection remains valid despite shortcomings in the genotyping method, because it was performed with the same initial population. Our finding that only about one third of the top 5% genes was jointly associated with both DENV-1 resistance and DENV-3 resistance cannot be explained by differences in genetic ascertainment because for both DENV types the genes were identified with the same set of SNPs in the same population.

The bulk of significant SNPs and genes were found on chromosome 1, particularly around the centromere. This observation could be due to at least two, non-mutually exclusive explanations. First, most of the genetic loci underlying DENV susceptibility in the Bakoumba population could be truly located on chromosome 1. QTL associated with DENV susceptibility in *Ae. aegypti* were previously detected on all three chromosomes [15, 17, 47, 48], and the genetic architecture of pathogen susceptibility is known to vary across host populations [49]. Second, the centromeric region of chromosome 1 contains the M locus responsible for sex determination in *Ae. aegypti* [17, 50]. We previously reported that the M locus is surrounded by a large region of reduced recombination between the sex chromosomes, which is associated with sex-specific genetic differentiation and elevated linkage disequilibrium (LD) over about 100 Mb in natural *Ae. aegypti* populations [51]. Because statistical power to detect genotype-phenotype associations increases with LD, it is possible that elevated LD may have inflated the proportion of significant genotype-phenotype associations in the centromeric region of chromosome 1. Nevertheless, as mentioned above, comparisons using the same population and the same markers remain valid because any genotyping bias should be consistent across experimental conditions.

The candidate genes that we identified deserve further investigation at the functional level. Although a genetic association does not necessarily imply differential expression, it would be interesting to determine whether candidate genes also display allelic differences in gene expression. Gene knockdown assays could be used to confirm the functional implication of the candidate genes [52, 53], although gene knockouts would provide a more definitive answer [54] because they are less prone to false negatives. Mosquito genes uniquely associated with resistance to DENV-3 infection were enriched in genes with zinc ion binding activity, whereas genes associated with resistance to both DENV-1 and DENV-3 infection were enriched in genes with ATP binding activity and sulfur compound transmembrane transporter activity. Zinc is an essential cofactor that ensures the proper folding and functioning of not only cellular proteins but also viral proteins [55]. To our knowledge, there is no prior evidence for a link between zinc ion binding activity and mosquito-virus interactions, however host cellular systems controlling zinc balance are known to interfere with virus replication [55]. Likewise, ATP binding activity and sulfur compound transmembrane transporter activity have not been specifically reported to participate in mosquito-virus interactions, however the high dependence of viruses on the cellular machinery makes any molecular function potentially relevant to host-virus interactions.

Although the forces driving the evolution of DENV resistance in *Ae. aegypti* are largely unknown, one evolutionary implication of the distinct gene set underlying resistance to DENV-1 and DENV-3 is that the evolution of resistance to one DENV type is not expected to lead to the correlated evolution of resistance to another DENV type. This finding is consistent with the absence of virus cross-resistance [56] and the lack of genetic trade-offs between the levels of resistance to different viral genotypes [57] in *Drosophila*.

The epidemiological relevance of our results is difficult to assess because dengue epidemiology is poorly documented in Gabon. A recent study reported DENV-3 circulation in 2016–2017 [58] whereas previous dengue outbreaks were mainly associated with DENV-2 in 2007 and in 2010 [59], but this information is insufficient to make a link between the DENV-1

resistance phenotype of the Bakoumba population and the relative lack of this DENV type in recently reported outbreaks in Gabon. In our experiments, we used a DENV-1 isolate from Thailand and a DENV-3 isolate from Gabon but the geographical origin of the virus is unlikely to have influenced the results due to the lack of evidence for DENV local adaptation to *Ae. aegypti* populations [60].

In conclusion, our results support a model where the genetic basis of resistance to DENV infection in *Ae. aegypti* has a "universal" component that acts across types and strains, and a type- or strain-specific component [15–17]. In the Bakoumba population, the DENV type-specific component of resistance consists, at least in part, of a distinct set of genes. Our gene enrichment analysis did not identify a functional overlap between the genes uniquely associated with DENV-1 resistance, uniquely associated with DENV-3 resistance, or associated with both. We speculate that *Ae. aegypti* resistance to DENV infection may involve different pathways and different molecular mechanisms. Probing this hypothesis will require follow-up studies to functionally test the effect of candidate genes by reverse genetics.

## Materials and methods

### Ethics statement

The Institut Pasteur animal facility has received accreditation from the French Ministry of Agriculture to perform experiments on live animals in compliance with the French and European regulations on care and protection of laboratory animals. This study was approved by the Institutional Animal Care and Use Committee at Institut Pasteur under protocol number 2015–0032.

### Cells and virus isolates

C6/36 cells (derived from *Ae. albopictus*) were cultured in Leibovitz's L-15 medium (Life Technologies) supplemented with 10% fetal bovine serum (FBS, Life Technologies), 1% non-essential amino acids (Life Technologies) and 0.1% Penicillin-Streptomycin (Life Technologies) at 28˚C. DENV-1 isolate KDH0026A was originally obtained in 2010 from the serum of a dengue patient in Kamphaeng Phet, Thailand [15]. DENV-3 isolate GA28-7 was originally obtained in 2010 from the serum of a dengue patient in Moanda, Gabon [59]. Informed consent of the patient was not necessary because viruses isolated in laboratory cell culture are no longer considered human samples. Virus stock was prepared using C6/36 cells, and viral infectious titers were measured on C6/36 cells using a standard focus-forming assay (FFA) as previously described [61].

### Mosquito rearing and exposure to DENV

All experiments were carried out with *Ae. aegypti* mosquito colonies derived from wild populations sampled in Bakoumba, Gabon in 2014, and Cairns, Australia in 2013. Both colonies had been maintained in the laboratory for 8 generations at the time of the experiments. Mosquitoes were reared and exposed to DENV as described previously [61]. Briefly, mosquitoes were reared under standard insectary conditions (28˚C ± 1˚C, 75 ± 5% relative humidity, 12:12 hour light-dark cycle) with a diet of fish food for larvae (Tetramin), and a 10% sucrose solution for adults. Twenty-four hours before experimental infection with DENV, 5- to 7-day-old female mosquitoes were deprived of sucrose solution. Fresh rabbit erythrocytes were washed and supplemented with ATP at a final concentration of 10 mM to stimulate blood uptake by mosquitoes. The blood meal consisted of a 2:1 mixture of erythrocytes and virus suspension, and was provided for 15 minutes using an artificial membrane feeding system

(Hemotek) with pig intestine as a membrane. To expose mosquitoes to different virus concentrations in the dose-response experiments, virus titer in the blood meal was adjusted prior to its preparation by diluting the virus stock in cell culture medium. Aliquots of the blood meal were collected prior to feeding, and viral titers subsequently determined by FFA. Blood-fed females were transferred to 1-pint carton boxes, and kept under controlled conditions (28˚C ± 1˚C, 70 ± 5% relative humidity, 12:12 hour light-dark cycle) with access to a 10% sucrose solution.

## DENV RNA detection

Mosquito bodies (abdomen and thorax) were ground for two rounds of 30 seconds at 5,000 rpm in a mixer mill (Precellys 24, Bertin Technologies) in 300 μL of squash buffer (10 mM Tris-Hcl pH 8.0, 50 mM NaCl, 1.25 mM EDTA diluted in Ultrapure water) supplemented with Proteinase K (Eurobio-Ingen) at a final concentration of 0.35 g/L. Lysate was heated to 56˚C for 5 minutes, and 98˚C for 10 minutes. Complementary DNA was synthetized using random hexamers and the M-MLV reverse transcriptase (Life Technologies). Five μL of lysate were included in a 20-μL reaction following the manufacturer's instructions. Finally, a PCR assay using DreamTaq polymerase (Life Technologies), and following manufacturer's instructions was used to generate amplicons from the conserved *NS5* region of DENV-1 (forward 5'-CGAAGATCACTGGTTCAGCA-3'; reverse 5'-ACATCCATCACGGTTCCATT-3') and DENV-3 (forward 5'-AGAAGGAGAAGGACTGCACA-3'; reverse 5'-ATTCTTGTGTCC CAACCGGCT-3'). Representative pictures of the electrophoresis gels and a rationale for scoring RT-PCR readouts are presented in S2 Fig.

## Virus titration

Virus titration was performed by FFA modified from a published protocol [62], as previously described [61]. In brief, a 96-well plate was seeded sub-confluently with C6/36 cells, inoculated with the viral suspension, and covered with an overlay medium containing 1.6% carboxyl methylcellulose solution. After 5 days of incubation at 28˚C, cells were fixed using 3.6% formaldehyde. Virus staining was performed using a primary mouse anti-DENV complex monoclonal antibody (MAB8705, Merck Millipore), and a secondary Alexa Fluor 488-conjugated goat anti-mouse antibody (Life Technologies). Infectious foci were counted using a fluorescent microscope and focus-forming units/mL (FFU/mL) subsequently determined.

## DNA extraction

DNA was extracted from individual mosquito heads using a protocol originating from Dr. Pat Roman's laboratory, and adapted from a previous publication [63]. First, the mosquito head was ground during two rounds of 30 seconds at 5,000 rpm in a mixer mill (Precellys 24, Bertin Technologies) in 300 μL of Pat Roman's buffer (0.1 M NaCl, 0.2 M sucrose, 0.1 M Tris-Hcl pH 8.0, 0.05 M EDTA pH 8.0, 0.5% SDS, adjusted at pH 9.2). Lysate was incubated at 65˚C for 35 minutes. While still warm, 42.5 μL of 8 M potassium acetate was added to each tube, followed by gentle mixing. After 30 minutes of incubation on ice, lysate was transferred to a new tube and centrifugated for 15 minutes at 12,000 $g$. Supernatant was transferred to a fresh tube, and mixed with 500 μL of 96% ethanol. DNA was left to precipitate overnight at -20˚C. Then, tubes were centrifugated for 30 minutes at 12,000 $g$. The pellet was washed once with ice-cold 75% ethanol, and then with ice-cold 96% ethanol, each time followed by 5-minute centrifugation at 12,000 $g$. Finally, the pellet was resuspended in 30 μL of TE buffer (10 mM Tris-HCl pH 8.0, 1 mM EDTA pH 8.0).

## Library preparation and sequencing

The amount of DNA isolated from each head was quantified using the Picogreen (Thermo Fisher Scientific) reagent following the kit protocol. To prepare the 12 sequencing libraries (2 DENV types x 2 phenotypes x 3 replicate experiments), individual DNA samples were combined to obtain 1.0 ng of DNA per pool. Individual DNA concentrations were adjusted prior to pooling so that each individual contributed the same amount of DNA to the pool. Library preparation and exome capture were performed following the detailed Nimblegen SeqCap EZ Library Protocol version 5 provided by Nimblegen (Roche). Briefly, genomic DNA was sheared using a Covaris to around 350-bp fragments and libraries were prepared using the Kapa HTP/LTP (Roche) library preparation kit following kit instructions. At the end of library preparation, the DNA concentration was quantified by Qubit (Thermo Fisher Scientific) and equal amounts of DNA from each library were pooled and allowed to hybridize with the exome-capture probes for 20 hours. Probes targeting the coding DNA sequences (CDS) were designed and synthesized by Nimblegen/Roche following previous studies [34–36]. Probe design used the AaegL3 genome build of *Ae. aegypti* [28] as this was the latest version of the genome at the time of the study. Genome coordinates for probes are provided as the BED file in S1 File. The exome-enriched libraries were sequenced in paired end on an Illumina NextSeq 500 instrument with a high-output v2 kit 300 cycles (Illumina). Raw sequencing data were deposited to the European Nucleotide Archive repository under accession number PRJEB36810.

## SNP detection

Sequencing reads were trimmed and cleaned of adaptor sequences using Trimmomatic [64] before mapping onto the AaegL5 genome assembly using BWA MEM [65, 66] The resulting BAM files were further sorted and indexed and PCR duplicate were removed using the SAMtools v1.2 sort, index and rmdup programs, respectively, with default options [67]. The 12 pooled-sequencing BAM files were processed using the mpileup program in SAMtools [67]. Variant calling was performed on the resulting mpileup file using Varscan v2.3.6 [68] with default options except for the minimum coverage threshold (-min-coverage option) set to 10 reads, the variant minor allele frequency threshold (-min-var-freq option) set to 0.01, and the average base quality score threshold (-min-avg-qual option) set to 20. The resulting VCF file was subsequently processed with custom awk scripts to retain only bi-allelic SNPs covered by more than 20 reads and less than 400 reads in each pool. In total, 1,498,923 SNPs were available for association testing with a median coverage ranging from 85X to 149X per pool.

## Association testing

Exome-wide association testing of the 12 phenotypic pools was performed using the $C_2$ contrast statistic implemented in the program BayPass 2.2 [31, 32]. This statistic compares allele frequencies between two groups of pools (resistant vs. susceptible), while properly accounting for the overall correlation structure resulting from the shared origin of the pools. Of note, the model implemented in BayPass allows to properly account for the sampling of reads from the (unobserved) allele count data in pooled-sequencing data when estimating and comparing the (unobserved) allele frequencies. Here, the 12 pools originated from the same colony derived from individuals sampled in Bakoumba, Gabon. To improve the statistical power of association tests, a diagonal structure was assumed for the scaled covariance matrix of pool allele frequencies as in the model initially proposed by Nicholson *et al.* [69]. The elements of the diagonal matrix were estimated as described in Gautier *et al.* [70] using the option *-nicholson-prior* newly introduced in version 2.2 of the BayPass program [31]. They can be interpreted as

the inverse of the actual effective sample size of the pools and ranged from 2.50% to 14.7% (with a median equal to 3.62%). Three independent runs of BayPass (using the option -seed) were performed to estimate the contrasts for each SNP. The estimated $C_2$ values together with the model hyper-parameters were highly correlated across the different runs (e.g., Pearson's correlation $>0.985$ for $C_2$ values) and only the results from the first BayPass run were retained. Support for association between each SNP and the DENV resistance phenotype was evaluated based on the $p$ values derived from the corresponding contrast statistic. The $p$-value distribution was close to uniform except for extremely small $p$ values. For each DENV type, gene-specific scores were calculated by adding the estimated $C_2$ values of their underlying SNPs. Gene positions were obtained using the base feature annotation file for the AaegL5 genome assembly (available at https://www.vectorbase.org/organisms/aedes-aegypti). Using custom awk scripts, a total of 964,398 SNPs (excluding SNPs with a minor allele frequency $<0.05$) were assigned to 10,797 different genes with a median number of 62 SNPs per gene (ranging from 1 to 2,377). When a SNP mapped to several genes, it was arbitrarily assigned to only the first ranking one. For each DENV type, support for association of the gene scores with the resistance phenotype was evaluated based on empirical $p$ values computed as follows. Let $k$ represent the number of SNPs mapping to a given gene $i$ and $n_k$ the number of loci of $k$ consecutive SNPs (including gene $i$) over the whole genome. Locus-specific scores were calculated as the sum of the $k$ $C_2$ values estimated for the SNPs underlying the $n_k$ loci. The empirical $p$ value of gene $i$ was computed as the proportion of the $n_k$ loci with a higher score than the gene $i$ score.

### Gene enrichment analysis

Genes with the highest scores were analyzed for functional enrichment by submitting the gene lists to the DAVID Knowledgebase v6.8 [71]. Enriched terms according to the *Ae*. *aegypti* background were clustered with REVIGO [72]. A treemap for gene ontology (GO) terms enriched in the molecular function category was generated using an in-house R script. In the treemap, each term is represented by a rectangle whose size is proportional to the number of genes in the list out of the total number of genes with similar roles in the *Ae*. *aegypti* genome. Terms with a semantic similarity, e.g. genes with a proximity in the GO graph below 0.7 of similarity, are represented in the same color and the centroid term is emphasized.

### Supporting information

**S1 Fig. SNPs associated with DENV-1 and DENV-3 infection in the Bakoumba population.** (**A**) Manhattan plot of $p$ values representing the association between SNP frequency and DENV-1 resistance, distributed along the three chromosomes. SNP frequency was compared between phenotypic pools of mosquitoes that are either resistant of susceptible to DENV-1. (**B**) Manhattan plot of $p$ values representing the association between SNP frequency and DENV-3 resistance, distributed along the three chromosomes. SNP frequency was compared between phenotypic pools of mosquitoes that are either resistant of susceptible to DENV-3. In (**A**) and (**B**), each dot represents a single SNP and is colored according to the statistical significance of the genotype-phenotype association. Blue dots represent the lower 5% of $p$ values and red dots represent the lower 2.5% $p$ values. (**C**) Venn diagram of the 0.001% most significant SNPs associated with DENV-1 infection, DENV-3 infection, or both.
(TIF)

**S2 Fig. Scoring of the mosquito body infection phenotype.** Photos of representative electrophoresis gels of RT-PCR products for DENV-1 (**A**) and DENV-3 (**B**) detection in mosquito bodies. The readout was based on five scores (exemplified in red font) as follows: 1 = clear and

bright band at the right height; 2 = clear and moderately bright band at the right height; 3 = weak band at the right height; 4 = one or several bands at an unexpected height (sometimes accompanied by the right band); 5 = no band. A sample was only considered DENV-positive when its score was 1 or 2. For DENV-1 all five scores were typically present on the gels, whereas for DENV-3 scores 3 and 4 were typically absent. The + and–symbols denote positive and negative controls, respectively.
(TIF)

**S1 Table. List of the 0.001% most significant SNPs associated with DENV-1 infection, DENV-3 infection, or both.**
(XLSX)

**S2 Table. List of the 5% most significant genes associated with DENV-1 infection, DENV-3 infection, or both.**
(XLSX)

**S1 File. Exome capture probe design.**
(ZIP)

## Acknowledgments

We are grateful to Gordana Rašić and Ary Hoffmann for providing the mosquito colony from Cairns. We thank Catherine Lallemand for assistance with mosquito rearing and three anonymous reviewers for helpful comments on an earlier version of the manuscript.

## Author Contributions

**Conceptualization:** Laura B. Dickson, Sarah H. Merkling, Albin Fontaine, Louis Lambrechts.

**Data curation:** Laura B. Dickson, Sarah H. Merkling, Mathieu Gautier, Amine Ghozlane.

**Formal analysis:** Laura B. Dickson, Mathieu Gautier, Amine Ghozlane, Louis Lambrechts.

**Funding acquisition:** Louis Lambrechts.

**Investigation:** Laura B. Dickson, Sarah H. Merkling, Davy Jiolle, Christophe Paupy, Diego Ayala, Isabelle Moltini-Conclois, Albin Fontaine.

**Methodology:** Mathieu Gautier, Louis Lambrechts.

**Supervision:** Louis Lambrechts.

**Visualization:** Laura B. Dickson, Sarah H. Merkling, Amine Ghozlane, Louis Lambrechts.

**Writing – original draft:** Laura B. Dickson, Louis Lambrechts.

**Writing – review & editing:** Laura B. Dickson, Sarah H. Merkling, Mathieu Gautier, Amine Ghozlane, Davy Jiolle, Christophe Paupy, Diego Ayala, Isabelle Moltini-Conclois, Albin Fontaine, Louis Lambrechts.

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
