## [Decision Letter · Decision Letter 0]

30 Mar 2020

Dear Dr Lambrechts,

Thank you very much for submitting your Research Article entitled 'Exome-wide association study reveals largely distinct gene sets underlying specific resistance to dengue virus type 1 and 3 in Aedes aegypti' to PLOS Genetics. Your manuscript was fully evaluated at the editorial level and by independent peer reviewers. The reviewers appreciated the attention to an important problem, but raised some substantial concerns about the current manuscript. Based on the reviews, we will not be able to accept this version of the manuscript, but we would be willing to review again a much-revised version. We cannot, of course, promise publication at that time.

If you decide to revise the manuscript for further consideration at PLOS Genetics, please aim to resubmit within the next 60 days, unless it will take extra time to address the concerns of the reviewers, in which case we would appreciate an expected resubmission date by email to plosgenetics@plos.org.

[LINK]

We are sorry that we cannot be more positive about your manuscript at this stage. Please do not hesitate to contact us if you have any concerns or questions.

Yours sincerely,

Giorgio Sirugo

Associate Editor

PLOS Genetics

Gregory P. Copenhaver

Editor-in-Chief

PLOS Genetics

Reviewer's Responses to Questions

**Comments to the Authors:**

Reviewer #1: The study by Dickson and colleagues report that different sets of genes condition the refractory phenotype of Aedes aegypti to dengue virus (DENV) serotype 1 and 3. The authors conducted exome sequencing of mosquito pools showing differential interaction with the two serotypes, and applied a SNP based association statistics to predict the host genes that likely controlled the observed phenotypes. I find the study interesting but highly intriguing. Here are the problems of this study-

Major Issues

1. The central hypothesis of this study is that if a vector population responds differentially to two serotypes, then factors associated with the host could explain that phenotypic variation (lines 165-167). But, the observed resistance phenotype could also be due to factors associated with the virus. What was done to rule out that possibility?

2. The major premise of this study also lack the very fact that genetic changes in the non-coding regions of genome are major contributors of gene regulation. It would make better sense if the authors had sequenced the whole genome, not the exome alone, to identify the associated SNPs. To me, this is a major flaw in the experimental design relative to the said objective of the study.

3. The authors identified two sets of non-overlapping genes that condition resistance exclusively to DENV1 vs DENV3. To accept that result, it is necessary to provide data that those genes differentially respond to infection with DENV1 vs DENV3. To show that the authors need to provide the expression level of genes, may be in the dissected midguts of the females, after infecting with the two serotypes.

Minor issue

The discussion is largely descriptive. The authors should explain what is the biological meaning of their findings. What are the functional and evolutionary implication that a vector mosquito must utilize non-overlapping gene sets to defend infection by DENV serotypes. Is there an evolutionary benefit for the vector? Is it relevant (the host effect) to differential prevalence of DENV1 relative to DENV3?

Reviewer #2: Here, the authors have produced a paper investigating the genetic architecture of dengue virus (DENV) resistance in a population of Aedes aegpti from Bakoumba, Gabon. This population displays a stronger resistance phenotype to DENV-1 compared to DENV-3. The authors used experimental bloodmeal exposures and exome sequencing of large phenotypic pools that were either susceptible or resistant. The paper is well written but I believe there are several things that need to be addressed to facilitate understanding by the reader. First, the authors demonstrate that their Bakoumba population of Ae. aegypti have differential susceptibility to DENV-1 vs. DENV-3 and that this is likely the result of mosquito genetic factors because a population of Ae. aegypti from Cairns, Australia were equally susceptible to DENV-1 and DENV-3. While I do not expect the authors to do additional experiments, I do wonder why there was no comparison between the two mosquito populations? This seems like an ideal way to validate susceptibility loci in the Bakoumba population. Second, it would be informative to know DENV epidemiological data for Bakoumba, Gabon. Critically, have there been outbreaks of DENV-1? The authors used a DENV-1 isolate from Thailand for experiments, whereas the DENV-3 isolate was from Gabon. This is somewhat controlled with the Cairns vector competence experiments, but it does raise the question of the appropriateness of the DENV-1 isolate for GxG experiments. Third, as I understand it, exome sequencing was done from mosquito heads only. Given the tissue-specific nature of resistance mechanisms in different host-pathogen combinations, is there potential for information to be missed when not using the whole body?

Reviewer #3: This is a very interesting and well-written paper investigating how differential SNP frequencies in the exome of an Aedes aegypti population from Gabon (affecting different gene sets) account for variable resistance levels to dengue 1 and dengue 3 viruses. The mosquito population is significantly more resistant to DENV1 than to DENV3. Mosquitoes were exposed to various doses of either virus followed by selection for extremely DENV3 susceptible/resistant versus DENV1 susceptible/resistant phenotypes. Once selected, they were subjected to whole-exome sequencing. Based on computational differential SNP analysis, the authors revealed that resistance to DENV1 in the Gabon population was largely based on different gene sets than resistance to DENV3.

This work is important as it shows on a global level how gene polymorphisms in Ae. aegypti, an organism with a highly complex genome structure, contribute to pathogen resistance phenotypes. The experimental design has been carefully chosen and the data analysis looks thorough.

There are a few important issues that need to be addressed.

Abstract:

Line 38: ….the exomes of…..

Introduction:

Line 96: …population in the long term….

Line 114: how do you define here strongly resistant versus moderately resistant?

Line 126: …gene-wide….

Results:

The Results section should be subdivided using sub-headers for the different paragraphs.

Line 189 onwards: This is not very clear to the reader - what does ‘individually phenotyped’ mean, testing of carcasses via RT-PCR for the presence/absence of virus?? “DNA was extracted from 182, 174, 176 females…..(carcasses/head tissues??); …..combined into 12 standardized phenotypic pools of 30-48 individuals.” What does standardized mean here? What individuals? This all should be made more clear using precise descriptions because this information is crucial to understand the experimental design und ultimately the results.

Table 3: “…tested DENV-positive or DENV-negative…”perhaps add here “based on RT-PCR results”

Line 218: is there a reason for using 2 different genome assembly/annotation versions? The analysis still could have been done using version 3 instead of 5? Would this likely/possibly affect the results? A comment here would be helpful.

Additional comment: since virus resistance levels of the “exosome samples” were assessed by RT-PCR, perhaps it would be a good idea to include a representative gel image into the supplemental information showing various PCR amplicons for DENV1 and DENV3 and also positive/negative controls. When a sample was considered negative, was there absolutely no signal at all? What about a very faint signal? How was this classified?

Discussion:

Perhaps also add some discussion points about possible functions of genes with zinc binding activity and those with ATP binding activity / sulfur compound transmembrane transporter activity in the context of mosquito infection with DENV.

Line 288:….about 2% of the entire……

Line 301: …in genetic ascertainment because for…. (delete ‘in’).

Line 305: change “Thus” for “This”

Line 312: ….surrounded by a large region…..

Materials and Methods:

Describe, how low and high DENV doses were prepared, validated, and administered to mosquitoes.

Line 345 & 347: replace “derived” for “obtained”

Line 375: …..was synthetized using random hexamers…..5 ul of lysate was included in a 20 ul reaction, following…..

Line 412: this is not clear: “…..equal amounts of DNA from each individual were pooled to reach the required 1.0 ng of DNA required for library preparation.” Please re-phrase/clarify.

Figure legends:

Line 702:….at 10 days post….

Figures:

Figure 1 and experiment referring to figure 1: two separate experiments were conducted in which Gabon and Australia mosquitoes were exposed to 3 different infectious doses ranging from 104 to 107 FFU/ml using DENV1 and DENV3. This experimental setup is difficult to retrieve from Fig. 1A. Where/how are the 3 different infectious doses shown? Based on that description, 24 data points should be shown instead of those 22 data points being presented. This is confusing. Perhaps the graph should be separated into two graphs: one for experiment 1 and another one for experiment 2.

Furthermore, the legend, open closed circles for experiment 1 & 2 and large, small circles for different N are way too tiny.

Figure 2: perhaps it would be helpful to extent the figure showing a complete experimental flow chart including number of mosquito carcasses tested for virus titers (FFU/ml), number of replicates, number of head tissues collected, pooling strategy of head tissue samples, and information on sequencing/data analysis strategy.

**Have all data underlying the figures and results presented in the manuscript been provided?**

Reviewer #1: None

Reviewer #2: Yes

Reviewer #3: Yes

PLOS authors have the option to publish the peer review history of their article (what does this mean?). If published, this will include your full peer review and any attached files.

Reviewer #1: No

Reviewer #2: No

Reviewer #3: No

---

## [Decision Letter · Decision Letter 1]

23 Apr 2020

Dear Dr Lambrechts,

We are pleased to inform you that your manuscript entitled "Exome-wide association study reveals largely distinct gene sets underlying specific resistance to dengue virus type 1 and 3 in Aedes aegypti" has been editorially accepted for publication in PLOS Genetics. Congratulations!

Yours sincerely,

Giorgio Sirugo

Associate Editor

PLOS Genetics

Gregory P. Copenhaver

Editor-in-Chief

PLOS Genetics

Comments from the reviewers (if applicable):

Reviewer's Responses to Questions

**Comments to the Authors:**

Reviewer #1: The revision is satisfactory.

Reviewer #2: All of my concerns have been allayed. I believe this now acceptable for publication.

Reviewer #3: line 202: perhaps change "The body of.....was..." to plural: Bodies of ......were....

**Have all data underlying the figures and results presented in the manuscript been provided?**

Reviewer #1: Yes

Reviewer #2: Yes

Reviewer #3: Yes

PLOS authors have the option to publish the peer review history of their article (what does this mean?). If published, this will include your full peer review and any attached files.

Reviewer #1: No

Reviewer #2: No

Reviewer #3: No

**Data Deposition**

http://datadryad.org/submit?journalID=pgenetics&manu=PGENETICS-D-20-00228R1

**Press Queries**

---

## [Editor Report · Acceptance letter]

15 May 2020

PGENETICS-D-20-00228R1 

Exome-wide association study reveals largely distinct gene sets underlying specific resistance to dengue virus types 1 and 3 in Aedes aegypti 

Dear Dr Lambrechts, 

We are pleased to inform you that your manuscript entitled "Exome-wide association study reveals largely distinct gene sets underlying specific resistance to dengue virus types 1 and 3 in Aedes aegypti" has been formally accepted for publication in PLOS Genetics! Your manuscript is now with our production department and you will be notified of the publication date in due course.

With kind regards,

Jason Norris

PLOS Genetics

On behalf of:
